# Comparative analysis of social issues toward medical abortion using mifepristone in South Korea and the United States: Topic modeling and sentiment analysis

**Minoh Ko**[1,©], **Dong-Young Park**[2,©], **Jung Mi Oh**[3], **Yun-Kyoung Song**[2*]

**1** College of Pharmacy, Daegu Catholic University, Gyeongsan-si, Republic of Korea, **2** College of Pharmacy, The Catholic University of Korea, Bucheon-si, Republic of Korea, **3** College of Pharmacy and Research Institute of Pharmaceutical Sciences, Seoul National University, Seoul, Republic of Korea

© These authors contributed equally to this work.
* yksong@catholic.ac.kr

## Abstract

This study aimed to compare social issues influencing the use of mifepristone for medical abortion between South Korea and the United States by analyzing media coverage through text mining techniques. News articles published between 2006 and 2022 related to mifepristone were collected using web scraping from representative news platforms in both countries. The collected data underwent preprocessing, followed by text analysis using the bag-of-words model. Topic modeling was conducted through Latent Dirichlet Allocation, and sentiment analysis involved using the lexicon-based methods specific to Korean and English texts. Analysis of 7,938 Korean and 650 United States articles revealed distinct patterns in media discourse on mifepristone and medical abortion. The United States coverage was polarized, encompassing clinical, legal, and access-related debates, whereas Korean articles prioritized access barriers, safety concerns, and the urgency for domestic approval, reflecting predominantly negative sentiments related to cultural stigma. This study highlights differences in the media portrayals and public perceptions of mifepristone between South Korea and the United States, underscoring the influence of regulatory contexts, public concerns and cultural attitudes on medical abortion discourse.

## Introduction

Abortion remains a socially divisive issue, balancing women's reproductive rights against moral and ethical opposition [1]. Access to abortion services varies globally depending on legislation, social values, and healthcare policies. Medical abortion using mifepristone has raised ongoing debates regarding its moral legitimacy and potential health risks, although often described as a less invasive alternative to surgery [2]. Approved by the United States (US) Food and Drug Administration (FDA) in

**Data availability statement:** All relevant data are within the manuscript.

**Funding:** This research was supported by the Basic Science Research Program through the National Research Foundation of Korea (NRF), funded by the Ministry of Education (NRF-2022R1C1C1011730), and by the BK21 FOUR-Advanced Program for SmartPharma Leaders, The Catholic University of Korea. There was no additional external funding received for this study.

**Competing interests:** The authors have declared that no competing interests exist.

2000 after several decades of use in European countries such as France, the United Kingdom. Italy and Sweden, mifepristone is now available in numerous countries, following the World Health Organization's (WHO) endorsement of its use in combination with misoprostol as an early abortion regimen [3–6]. Prevalence of medical abortion has increased worldwide, comprising more than half of all abortions in high-income countries. In the US, despite the FDA's regulatory oversight and the adoption of Risk Evaluation and Mitigation Strategy, reports of adverse events such as severe bleeding, ectopic pregnancy, and systemic infections persist [7–11]. In contrast, South Korea has yet to approve mifepristone despite the 2019 Constitutional Court ruling decriminalizing abortion up to 14 weeks of gestation [12,13]. Cultural stigma, conservative values, and limited public discourse on reproductive ethics continue to restrict access, underscoring the need for cautious, evidence-based policymaking than balances women's health with broader moral and societal consideration [13].

Media discourse plays a pivotal role in shaping public attitudes and policy debates on contested topics such as abortion [14,15]. Coverage of medical abortion and mifepristone reflects prevailing societal attitudes, cultural beliefs, and political contexts. Comparative analyses of news content across countries can illuminate how differing moral climates and policy priorities generate divergent narratives [16,17]. Such analysis can elucidate underlying factors that contribute to disparate approaches towards medical abortion, thereby supporting the evidence-based communication and policy strategies. In the digital era, computational text-mining techniques such as topic modelling and sentiment analysis enable systemic exploration of dominant themes, sentiment trends, and linguistic patterns in media representations of health issues [18]. However, limited research has compared media discourse on medical abortion across countries with contrasting regulatory and cultural contexts, notably South Korea and the US. Understanding these differences is crucial to evaluating how societal values and policy frameworks shape media narratives, thereby informing culturally sensitive and empirically grounded reproductive health policymaking.

Therefore, this study aims to comprehensively compare news media coverage in South Korea and the US through topic modeling and sentiment analysis, identifying predominant themes and sentiment patterns surrounding medical abortion with mifepristone. By clarifying how distinct national contexts influence media narratives and public perceptions, this research seeks to provide critical insights to guide policymakers, healthcare providers, and advocacy groups in developing culturally informed reproductive health policies.

## Methods

This study was a retrospective comparative analysis of published news articles using a text-mining approach. As the study was conducted using publicly available online news articles without any non-identifiable data and did not involve human participants, no sensitive or confidential data were included in the dataset, and all analyses were conducted on aggregated, anonymized text corpora. Therefore, it was exempted from ethical review by the Institutional Review Board (IRB) of Daegu Catholic University (IRB No. CUIRB-2022-E004). The study comprised three stages: (1)

data collection and preprocessing, (2) topic modeling using Latent Dirichlet Allocation (LDA), and (3) sentiment analysis using lexicon-based methods. The outline of the text analysis pipeline is described in Fig 1.

## Data collection and preprocessing

News articles related to mifepristone, published between January 1, 2006, and December 31, 2022, were collected from Google News [19] and Naver News [20] using web scraping techniques, the Python library: Beautifulsoup. The search included news articles that contained the keywords "Mifepristone," "Misoprostol," "Mifegyne," and "RU-486" in either the title or the main context. On the Korean news platforms these keywords were searched in Korean, while English terms were used for the US news platforms. On Google News, further targeted searches were conducted to retrieve relevant articles from various news media websites, such as British Broadcasting Corporation, the New York Times, and the Washington Post. To ensure the reliability and relevance of the collected articles, only articles from registered news media sources indexed by Google News and Naver News were included. Naver News was considered for South Korea owing to its dominant position in aggregating nearly all major domestic news outlets [21], and Google News was selected for the US because of its extensive and representative aggregation of leading American news sources [22]. Both platforms have news policies [23,24] to ensure that the content they provide comes from media outlets that adhere to best practices. Duplicate articles and those irrelevant to the study's context were identified and excluded through manual sorting and automated deduplication processes. There were no irrelevant articles identified after manual sorting. The collected articles underwent preprocessing, including data cleaning, stopword removal, and tokenization. Owing to the linguistic differences between Korean and English, language-specific preprocessing was performed. For Korean texts, morphological analysis was conducted using the Python natural language processing library 'Konlpy' and the Korean morphological analyzer 'Komoran.' Similarly, a Byte Pair Encoding-like method was used for out-of-vocabulary cases. English texts were pre-processed using the R packages 'stringr' and 'tm' for tasks such as converting words to lowercase; removing punctuation, numbers, and special characters; and eliminating stopwords. Additionally, Sublime Text 3 was used to convert multiple spaces into single spaces. Tokenization of Korean texts primarily focused on nouns. Textual features were extracted using the bag-of-words (BoW) model, and term frequency-inverse document frequency (TF-IDF) was calculated to determine the relative significance of words within articles [25]. The correlation between two words (X and Y) was quantified using the Phi coefficient ($\varnothing$), calculated as follows:

$$\varnothing = \frac{(ad - bc)}{\sqrt{(a+b)(c+d)(a+c)(b+d)}}$$

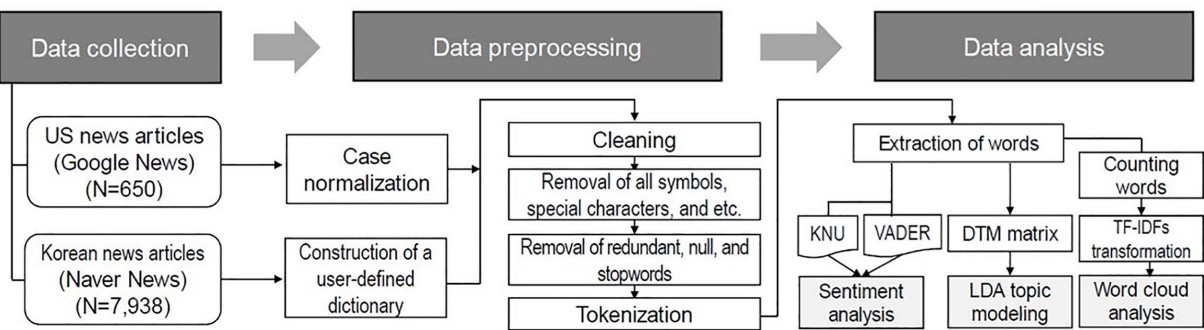

**Fig 1. Pipeline of the text analysis.** Abbreviations: DTM, document term matrix; KNU, KNU Korean sentiment lexicon; LDA, latent Dirichlet allocation; TF-IDF, term frequency-inverse document frequency; VADER, Valence Aware Dictionary and sEntiment Reasoner.

where:

 *a*: the number of cases where words X and Y are present,

 *b*: the number of cases with only word X,

 *c*: the number of cases with only word Y,

 *d*: the number of cases without word X or Y.

Finally, tokenization was conducted using 'Konlpy,' and feature extraction was performed using the 'Counter' function from Python's 'Collections' package and 'TfidfVectorizer' from the 'Scikit-Learn' library. All analyses were conducted in a Python 3.8 environment.

## Topic modeling

After preprocessing, topic modeling was performed using a common algorithm, the LDA model. The optional number of topics was determined based on performance indicators suggested by Griffiths in 2004 [26]. Gibbs sampling was used to estimate topic and keyword distributions. The allocation probability of each word being assigned to a specific topic was calculated using a Markov Chain Monte Carlo method. Topics were randomly assigned initially and iteratively updated using the following equation [26]:

$$P(z_{ij} = k \mid z_{-ij}, w) = \frac{(n_{d_i,k}^{-ij} + \alpha)(n_{k,w_{ij}}^{-ij} + \beta)}{(n_{d_i}^{-ij} + K\alpha)(n_k^{-ij} + V\beta)}$$

where:

 *k*: number of topics

 *V*: size of word set

 $w_{ij}$: the $j^{th}$ word of the document $d_i$

 $z_{ij}$: the subject of the $j^{th}$ word in the document $d_i$

 $\alpha$: parameters of the Dirichlet prior distribution of the document-topic distribution

 $\beta$: parameters of the Dirichlet prior distribution of the topic-word distribution

 $n_{d_i,k}^{-ij}$: number of words assigned to topic $k$ in the document $d_i$ (excluding current word)

 $n_{d_i}^{-ij}$: number of words assigned to all topics in the document $d_i$ (excluding current word)

 $n_{k,w_{ij}}^{-ij}$: number of words $w_{ij}$ assigned to topic $k$ (excluding current word)

 $n_k^{-ij}$: number of all words assigned to topic $k$ (excluding the current word)

Bootstrap iteration analysis was conducted to identify latent themes within topics. The LDA analysis was conducted using the Gensim library in a Python 3.8 environment [27].

## Sentiment analysis

Sentiment evaluation was conducted for each word or phrase in the dataset. For English texts, sentiment scores were assigned using the National Research Council Canada (NRC) Emotion Lexicon [28], which categorizes words by emotional associations. For Korean texts, the KNU Korean sentiment lexicon [29] was used. Additionally, the Valence Aware Dictionary and sEntiment Reasoner was applied to annotate the sentiment of each term and quantify sentiment intensity [30]. The log odds ratio (LOR) was calculated as a statistical measure of emotional content using the equation below, followed by normalization using a Min-Max scaler ranging from −1–1 [29]:

$$LOR = \log \frac{\frac{(Sentence_{positive})}{\sum (Sentence_{positive}+1)}}{\frac{(Sentence_{negative})}{\sum (Sentence_{negative}+1)}}$$

This metric evaluates the relative odds of positive versus negative sentiment within texts, with a high LOR signifying a strong positive sentiment and a low LOR indicating a strong negative sentiment. All statistical analyses were performed using R (version 4.3.1; Foundation for Statistical Computing, Vienna, Austria).

## Results

### Data description of the newspaper articles on mifepristone

Following the text analysis pipeline described in Fig 1, duplicate articles were automatically removed using a pre-specified script, after which all remaining articles were manually reviewed; no further exclusions were required. A total of 7,938 Korean articles and 650 US articles on medical abortion with mifepristone, published between 2006 and 2022, were included in the final analysis. The analysis revealed differences in the volume and thematic focus of the media coverage between the two countries (Fig 2). The Korean dataset contained a larger number of articles, with a surge in publications in 2021, approximately tripling the total from the preceding 3 years. This increase coincided with significant events in Korea. Until 2020, there was a general decline in the number of published articles, except in 2017, when a spike occurred owing to the reassessment of abortion laws prompted by a petition to the Blue House [31]. This declining trend was reversed in 2021, driven by the abolition of the abortion laws in Korea and Hyundai Pharmaceutical™'s initial application for mifepristone approval [32]. Conversely, the US demonstrated a consistent upward trend in the volume of news articles over the years (Fig 2).

### Word frequency of the newspaper articles on mifepristone

After preprocessing, 2,580 mifepristone-related corpora were obtained. The BoW analysis, presented through word clouds, highlighted differences in frequently used terms between the two countries (Fig 3). Table 1 presents the TF-IDF for each term in both countries and indicates the relative importance of each word. In both countries, 'abortion' and 'mifepristone' were identified as the most frequently occurring words, with 'abortion' having the highest relative importance. In

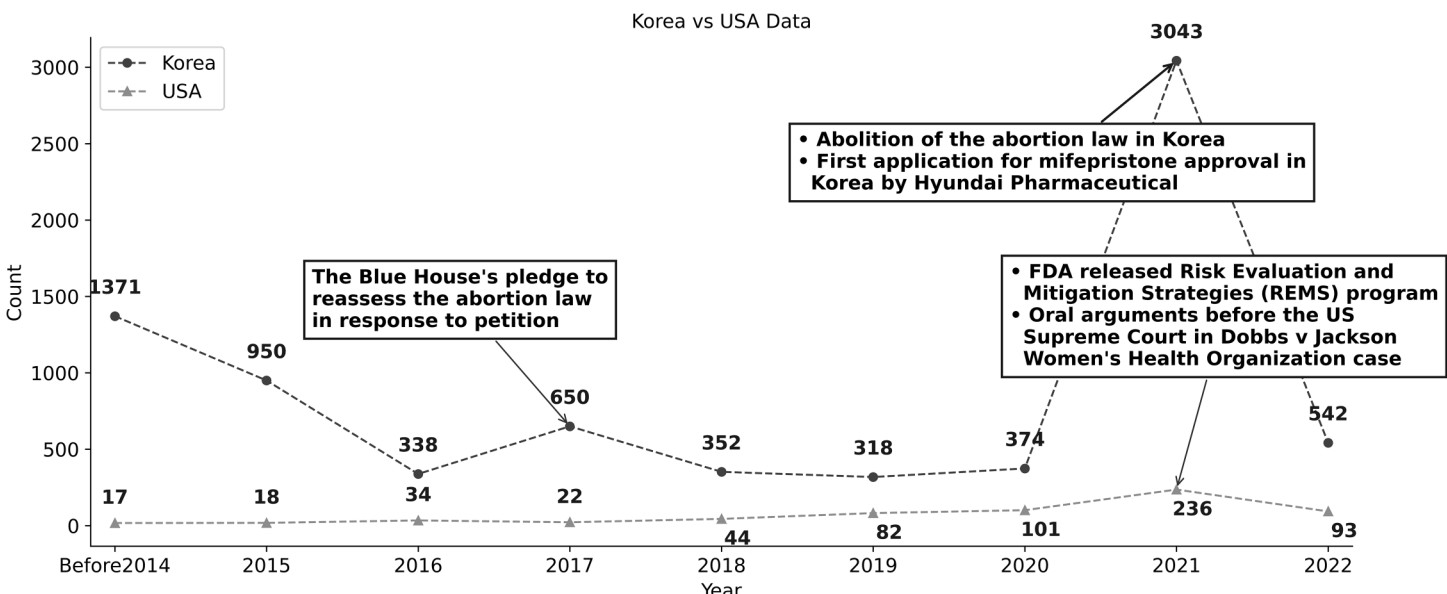

**Fig 2. Temporal distribution of the number of newspaper articles and major events in each country.** * News articles from South Korea were crawled from Naver News, and those from the United States (US) were obtained from Google News.

**(A)**

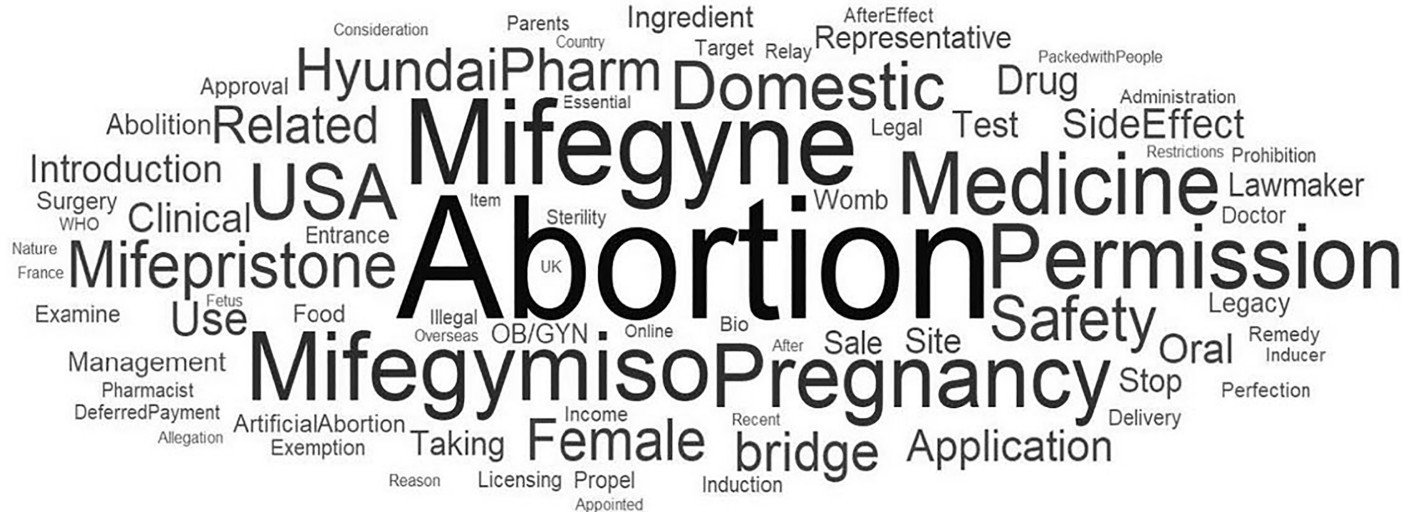

**(B)**

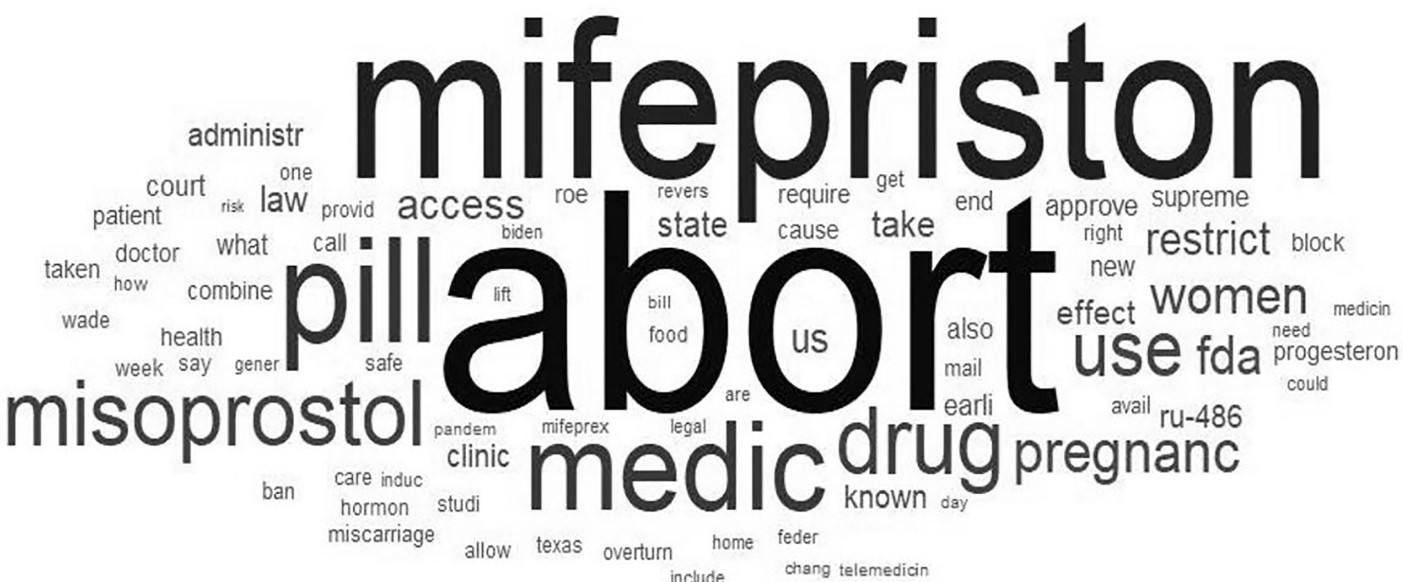

**Fig 3.** Word clouds related to mifepristone in the text-mining analysis of newspaper articles in (A) South Korea and (B) the US.

Korean articles, the prominent terms included 'approval', 'domestic,' and 'USA,' reflecting regulatory focus and comparative international perspectives. Safety-related terms such as 'side effects' and 'safety' also featured frequently. In contrast, the US dataset emphasized terms related to medication and access, such as 'pill', 'medication', 'drug', 'access', 'clinic', and 'use', indicating a stronger focus on practical and accessibility aspects of mifepristone. Furthermore, 'pregnancy' ranked third among Korean articles, following 'abortion' and 'Mifegyne,' whereas in the US, clinical terms such as 'clinic,'

**Table 1. Frequency of words related to mifepristone in the text-mining analysis of newspaper articles in (A) South Korea and (B) the United States (US) (top 20).**

| Rank | South Korea | | | The US | | |
|---|---|---|---|---|---|---|
| | Words | TF-IDF | N (%) | Words | TF-IDF | N (%) |
| 1 | Abortion | 494.6 | 6,881 (3.50) | Abortion | 50.3 | 870 (7.07) |
| 2 | Approval | 438.5 | 3,980 (2.02) | Mifepristone | 37.4 | 628 (5.10) |
| 3 | USA | 432.4 | 3,424 (1.74) | Pill | 34.1 | 91 (0.74) |
| 4 | Mifegyne® | 396.6 | 5,432 (2.76) | Medication | 31.6 | 340 (2.76) |
| 5 | Pregnancy | 357.4 | 3,997 (2.03) | Drug | 26.5 | 249 (2.02) |
| 6 | Domestic | 324.6 | 3,070 (1.56) | Misoprostol | 26.0 | 248 (2.01) |
| 7 | Medication | 318.8 | 3,065 (1.56) | Use | 23.9 | 202 (1.64) |
| 8 | Bridge | 317.7 | 2,386 (1.21) | Pregnancy | 19.5 | 154 (1.25) |
| 9 | Related | 316.1 | 2,260 (1.15) | Food and Drug Administration (FDA) | 19.4 | 132 (1.07) |
| 10 | Mifegymiso® | 305.6 | 3,240 (1.65) | Women | 17.1 | 137 (1.11) |
| 11 | Mifepristone | 300.9 | 2,793 (1.42) | Restriction | 15.2 | 111 (0.90) |
| 12 | Safety | 287.8 | 2,733 (1.39) | Access | 13.4 | 101 (0.82) |
| 13 | Women | 274.6 | 2,462 (1.25) | Take | 12.6 | 81 (0.66) |
| 14 | Hyundai Pharm Co., Ltd | 270.4 | 2,594 (1.32) | USA | 12.5 | 91 (0.74) |
| 15 | Clinical | 262.4 | 1,673 (0.85) | Known | 11.9 | 68 (0.55) |
| 16 | Utilization | 258.3 | 1,973 (1.00) | RU-486 | 11.7 | 69 (0.56) |
| 17 | Site | 257.4 | 1,436 (0.73) | Effect | 11.5 | 74 (0.60) |
| 18 | Trial | 253.8 | 1,487 (0.76) | Administration | 10.9 | 68 (0.55) |
| 19 | Application | 250.2 | 1,847 (0.94) | Approve | 10.7 | 59 (0.48) |
| 20 | Side effect | 238.6 | 1,714 (0.87) | Clinic | 10.6 | 66 (0.54) |

TF-IDF, term frequency-inverse document frequency.

'administration,' and 'use' were the most prominent. Additionally, terms such as 'bridging study' and 'Hyundai Pharm' were present in the Korean coverage due to the regulatory and commercial contexts.

## Topic modeling from the news articles on mifepristone

By applying the LDA method, seven distinct topics emerged for each country, highlighting thematic differences in the media discourse (Table 2). Korean articles focused on securing Asian distribution rights for Pictovir® (mifepristone) (21.1%), followed by discussions on safety concerns associated with medical abortion (15.7%). In contrast, US articles emphasized the legal frameworks underpinning medical abortion (26.2%) and its clinical applications (19.8%), reflecting broader societal and healthcare debates. Comparatively, while the distribution of topics within Korean articles presented relatively uniform probabilities (range of topic allocation rate, 11.9–21.1%), the US articles exhibited more variations in topic prevalence (range of topic allocation rate, 3.8–26.2%). Additionally, keyword analysis revealed that the US media frequently addressed clinical usage and practical administration of mifepristone, whereas the Korean coverage emphasized regulatory challenges and safety concerns.

## Sentiment analysis from the news articles on mifepristone

The sentiment analysis is illustrated using a graph plotting TF-IDF scores (x-axis) against normalized LOR (y-axis) (Fig 4). Notable differences emerged in sentiment patterns between the two datasets. English texts exhibited a more polarized sentiment pattern, with seven words categorized as neutral and one word each as distinctly positive and

**Table 2. Topics identified in newspaper articles related to mifepristone or Mifegyne® in (A) South Korea and (B) the US.**

| Topics | Keywords* | Topic allocation (%) |
|---|---|---|
| **(A) South Korea** | | |
| T1. Securing the Asian distribution rights for Pictovir® | USA, therapy, Pictovir®, permit, mifepristone, administer, secure | 21.1 |
| T2. Safety issues regarding medical abortion | USA, medication, surgery, side effect, safety, complete, aftereffect | 15.7 |
| T3. Various groups' reactions to the constitutional court's decision on the abortion law | related, representative, manage, chairman, risk, fetus, artificial abortion | 14.1 |
| T4. Issues raised on mifepristone by the parliament | abortion, trial, Mifegyne®, sales, medical doctor, import, congressman | 12.7 |
| T5. Need for the introduction of mifepristone | Mifegyne®, pregnancy, introduction, abortion, women, termination, miscarriage | 12.3 |
| T6. Illegal online trading of mifepristone | approval, mifepristone, site, pregnancy, abortion, Mifegyne®, progesterone | 12.2 |
| T7. Hyundai Pharmaceutical™'s application for Mifegymiso® approval | Mifegymiso®, approval, bridge, medication, Hyundai Pharmaceuitcal™, application, domestic | 11.9 |
| **(B) The US** | | |
| T1. Improving access to the abortion pill | abortion, education, pill, women, mifepristone, access, known | 11.7 |
| T2. Restrictions on medical abortion | FDA, abortion, mifepristone, pill, drug, restrictions, misoprostol | 18.5 |
| T3. Legal foundation for medical abortion | abortion, pill, mifepristone, education, Roe, misoprostol, women | 26.2 |
| T4. Supply of mifepristone for medical abortion | pharmacy, Hyundai Pharmaceutical™, Massachusetts, abortion, focus, upcoming, supply | 3.8 |
| T5. Utilization of mifepristone in the market | market, mifepristone, emergency, missed, pill, pregnancy, lower | 4.5 |
| T6. Clinical role of medical abortion | abortion, mifepristone, medical, pill, misoprostol, drug, clinic | 19.8 |
| T7. Mechanism of action of medical abortive agents | mifepristone, pill, abortion, misoprostol, pregnancy, drug, progesterone | 15.6 |

* English translation for Korean.

negative. Conversely, Korean texts revealed a broader sentiment distribution, with five words categorized as positive, one as neutral, and twelve as negative.

In Korean texts, words such as 'overseas' and 'approval' were associated with positive sentiment. In contrast, terms such as 'petition,' 'illegal,' and 'abolish' exhibited strongly negative sentiment, with normalized LOR values ranging from -0.75 to -1. The term 'permit' displayed a neutral sentiment (normalized LOR close to zero), while terms such as 'legalize' and 'sale' conveyed slight negativity. In English texts, sentiment polarization was more pronounced; 'treatment' demonstrated strong positivity (normalized LOR = 1), while 'restrictions' exhibited strong negativity (normalized LOR = 1). Additionally, the term 'women' held slightly negative sentiment in Korean articles but appeared neutral in the English corpus.

## Discussion

This study examines the social context of mifepristone in South Korea and the US through an analysis of news media discourse using medical informatics methodology. Leveraging analytical methods including text mining, topic modeling, and sentiment analysis, we identified prominent topics, underlying sentiments, and word associations that illustrate public attitudes and policy discourse regarding medical abortion within regulatory and healthcare contexts. The findings revealed media coverage in South Korea primarily emphasized the regulatory approval process, safety concerns, and social

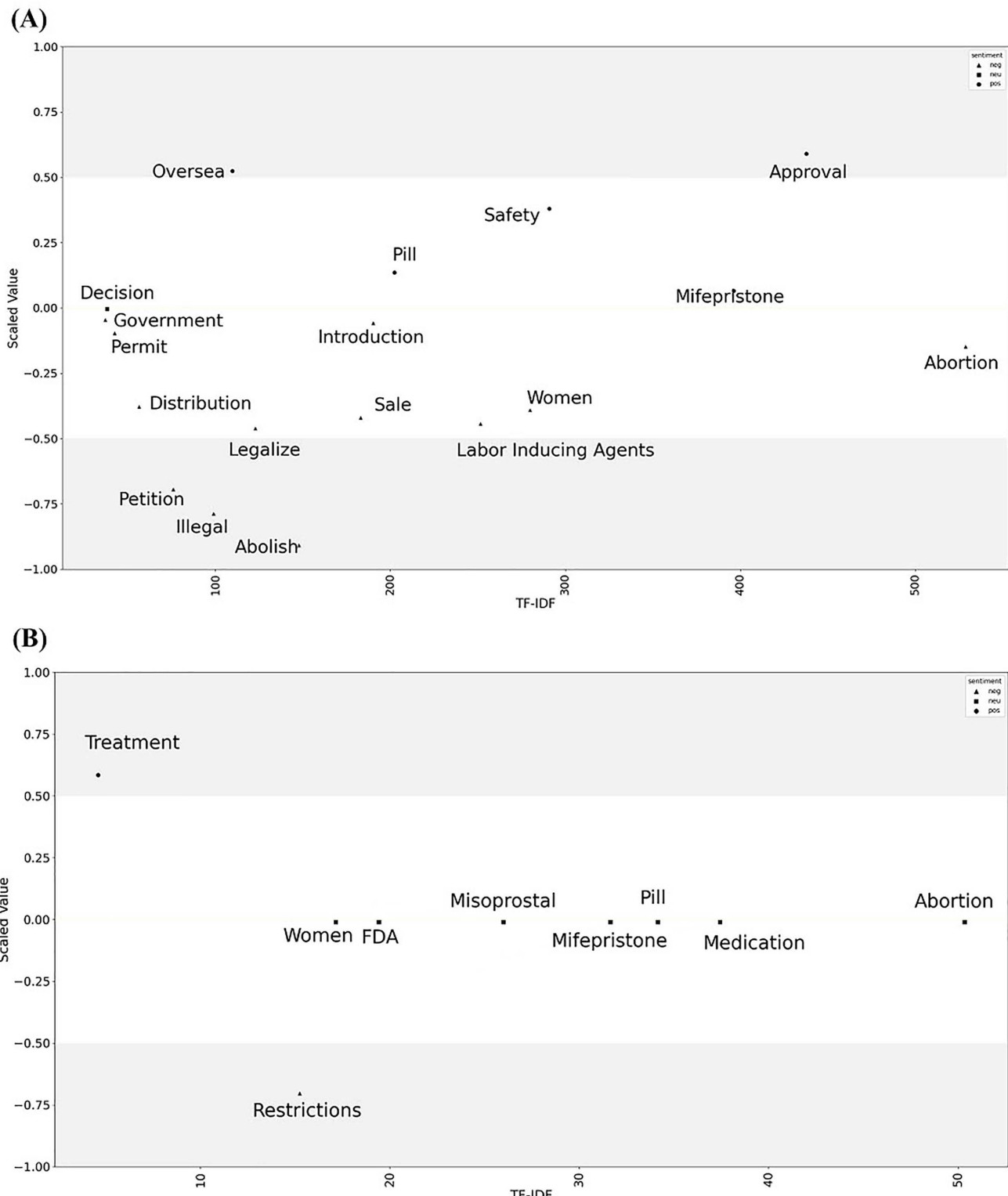

**Fig 4. Sentiment analysis based on the online newspapers related to mifepristone or Mifegyne® in (A) South Korea and (B) the US.**

   

controversies. In contrast, the US media discourse was characterized by legal debates, clinical applications, and accessibility, with more polarized sentiment patterns.

Our comparative analysis elucidated substantial differences in media portrayals and public discourses surrounding mifepristone between South Korea and the US. Notably, there was a disparity in the volume of news articles, as the South Korean dataset contained 7,938 articles vs. 650 articles for the US. This contrast can be attributable to the media attention in South Korea, largely driven by abolition of abortion restrictions in 2021 and the subsequent regulatory debates concerning mifepristone approval from Ministry of Food and Drug Safety. Although we cannot weigh the relative prominence of thematic content based on the article numbers, it is worth emphasizing that our primary goal was to provide qualitative overview and comparison of predominant themes and sentiments rather than conduct direct quantitative comparisons in publications between the two countries. Accordingly, we did not normalize the number of articles for the analysis. This decision stems from our view that the numerical contrast inherently represents distinct patterns of public interest and discourse across countries, and normalization could potentially introduce bias. Consequently, we are confident that this numerical imbalance will not substantially affect the interpretative validity of our thematic and sentiment analyses.

The findings from this study are meaningful within the context of recent policy reforms in both countries. In South Korea, the rescinding of restrictive abortion laws in 2021 marked a pivotal shift in reproductive rights [12,13,32], while in the US, Dobbs vs. Jackson Women's Health Organization case in 2022 [33] and the revised REMS framework of the FDA has facilitated safe access to mifepristone [9,10]. The results of our topic modelling captured press discussions of these regulatory shifts, offering insights into how public perceptions and discussions were shaped by these changes.

The TF-IDF analysis highlighted key differences in media coverage between South Korea and the US. Notably, the prominence of terms such as 'USA' and 'domestic' in Korean articles suggests an interest in comparative international practices. Additionally, variations in the usage of mifepristone's trade names—RU-486 in the US versus Mifegyne® or Mifegymiso® in South Korea—may indicate differences in adoption timelines and regulatory approaches between the two countries.

Furthermore, our study noted the illegal online trade of mifepristone frequently reported in Korean news coverage. Such reporting may suggest possible unintended associations between stringent regulatory controls and unsafe practices and misuse, raising public health and safety concerns [34]. These observations underscore the importance of developing balanced, evidence-based regulatory frameworks prioritizing public health and moral responsibility. Despite the role of legal changes in abortion debates, the word 'law' did not appear among the terms with high TF-IDF values in Korean articles. This omission may indicate a strong media focus on practical considerations related to the introduction of medical abortion rather than on legal debates surrounding the abolition of abortion laws. The media narratives are becoming increasingly important in shaping public belief and opinions, ultimately contributing to social change [14,35]. This practice may indicate the evolving nature of public discourse on abortion in South Korea, warranting further investigation into how media framing shapes the public understanding of complex health concerns.

Sentiment analysis on news coverage surrounding mifepristone in each country provided insights into the public perceptions. The US articles displayed a polarized pattern, with strongly positive sentiments associated with the term 'treatment' and negative sentiments linked to 'restrictions'. Other terms, such as 'women,' remained relatively neutral in the US dataset. In contrast, Korean articles demonstrated more varied sentiment distributions. Terms such as 'legalize', 'petition', and 'abolish' conveyed negative sentiments, reflecting both the restrictive environment and moral concerns surrounding abortion, while words like 'overseas,' 'treatment,' and 'safety' elicited positive sentiments. Notably, the term 'abortion' was associated with near-neutral sentiment in Korean texts, potentially indicating changing public attitudes and reflecting ongoing challenges in the societal acceptance of medical abortion in South Korea [13,32]. The neutrality associated with the word 'permit' and slight negativity with terms such as 'legalize' and 'sale' in Korean articles highlight the underlying ambivalence and moral unease within public discourse. Such sentiment patterns suggest persistent societal resistance and caution toward the acceptance of medical abortion, reflecting broader apprehensions about the moral and health implications

of expanding access to abortifacient drugs. These findings underscore the need for prudent regulatory approaches that prevent the normalization of medically and morally contentious practices. This contrasts with countries such as the US and Canada, where extensive evidence has contributed to widespread regulatory acceptance and clinical recognition of the effectiveness of mifepristone [36,37]. Furthermore, given the marginally neutral but somewhat negative connotation of the term 'labor-inducing agent', further large-scale cohort studies are warranted in countries where mifepristone is used, to assess its long-term safety in women. These findings emphasize the need for careful, value-sensitive and evidence-based policies that respect prevailing ethical perspectives while addressing emerging needs for safe clinical practices [38].

Topic modeling revealed differences in media emphases between the two countries. Korean articles focused on securing Asian distribution rights for Pictovir® (mifepristone) (21.1%), alongside discussions on medical abortion safety (15.7%). Conversely, the US articles prioritized the legal foundation of medical abortion (26.2%) and its clinical role (19.8%). These thematic differences resonate with the observations of Kulier et al. [8], regarding international variability in approaches toward medical abortion. Thus, our comparative analysis highlights the potential role of media framing in shaping public perceptions and policy orientations toward medical abortion.

This study had some limitations. While various big data open-sources are available for text mining, our analysis relied on web-crawled news articles from specific platforms and periods, which may not capture the full spectrum of media discourse on mifepristone. This study analyzed news articles to accurately capture formal media representations of regulatory, societal, and policy issues regarding medical abortion. Although leveraging social media content for the analysis could provide additional insights, individual opinions expressed through social media platforms usually introduce significant biases and distortions, precluding the accurate assessment of institutional discourse. We acknowledge a potential limitation regarding the effectiveness of lexicon-based sentiment analysis tools in accurately capturing the cultural nuances and contextual meanings inherent in Korean and English texts. While these tools have been validated and commonly used across diverse linguistic and cultural contexts in previous research [39–41], there remains a possibility of misinterpretation when applied to culturally distinct datasets. To mitigate it, the researchers involved in this study who are proficient in both languages cross-validated and any disagreements regarding contextual interpretation beyond automated sentiment analysis outputs were resolved through discussion. Nevertheless, our sentiment analysis results should be interpreted with caution, and future research should consider additional qualitative analyses or culturally customized lexicons to better account for language-specific subtleties. It should also be noted that the generalizability of these findings may be limited as media narratives and public sentiment towards medical abortion are influenced by the unique cultural, legal, and ethical environment.

Additionally, our analysis spans news articles between 2006 and 2022, potentially introducing a time-trend bias, as public discourse on media perspectives possibly evolved over this period. However, the study period (2006–2022) was specifically selected to capture critical legal and policy shifts related to abortion, enabling a thorough analysis of how media discourse adapted to these temporal shifts. This period saw global and regional regulatory milestones regarding mifepristone. In February 2006, the Federal Parliament of Australia overturned the Harradine Amendment—a legislative restriction that had previously limited the availability of mifepristone [42]. Moreover, in South Korea, the selected timeframe captured the 2017 Blue House petition demanding reconsideration of abortion regulations, the Constitutional Court's landmark 2019 ruling deeming the existing abortion ban unconstitutional, and the subsequent abolishment of abortion restrictions in January 2021 [12,13]. The observed increase in discourse following Hyundai Pharmaceutical's initial submission for mifepristone approval in 2021 further confirms that the analyzed period encompassed key events that likely influenced media narratives [32]. Thus, the extended analysis period adequately addressed potential concerns about overlooking temporal shifts in media discourse, illustrating how public narratives evolved alongside these landmark events. Future research could address these limitations by incorporating more news sources, employing dynamic topic modeling to capture temporal trends, and integrating qualitative methods.

Despite these challenges, the application of medical informatics methodologies in this study provides distinctive topics and sentiment patterns across countries. Specifically, Korean news coverage focuses more on safety issues and the

approval of mifepristone, whereas US articles exhibit a stronger emphasis on the broader concept of abortion itself. Even without directly quantifying causality, this study demonstrates the utility of health informatics in enhancing decision-making and policy formulation based on comprehensive, data-driven insights.

## Conclusions

This study identifies differences in media discourse on medical abortion using mifepristone between South Korea and the US, reflecting differences in the number of articles, key topics, and distinctive patterns of sentiments in each country. The Korean news coverage focused on regulatory challenges, safety concerns, and social perceptions, whereas the US coverage underscored clinical aspects and access-related debates. Our findings reveal differing public concerns and interests between the two countries, which may assist policymakers and healthcare providers in developing strategies to navigate complex societal dynamics surrounding medical abortion through media discourses. Furthermore, it is essential for the media to adopt more balanced and informative narratives when addressing culturally and ethically sensitive topics, including medical abortion.

## Acknowledgments

This manuscript was edited by Editage (www.editage.co.kr) for correction of grammatical errors, typos, and word selection. The authors also thank the anonymous reviewers for their constructive feedback on earlier drafts of this manuscript.

## Author contributions

**Conceptualization:** Dong-Young Park, Jung Mi Oh, Yun-Kyoung Song.

**Data curation:** Dong-Young Park.

**Formal analysis:** Dong-Young Park.

**Funding acquisition:** Yun-Kyoung Song.

**Methodology:** Minoh Ko, Dong-Young Park.

**Software:** Dong-Young Park.

**Supervision:** Yun-Kyoung Song.

**Validation:** Minoh Ko, Jung Mi Oh.

**Visualization:** Minoh Ko.

**Writing – original draft:** Minoh Ko.

**Writing – review & editing:** Dong-Young Park, Jung Mi Oh, Yun-Kyoung Song.

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
