## [Decision Letter · Decision Letter 0]

2 Jun 2025

Dear Dr. Song,

Thank you for submitting your manuscript to PLOS ONE. After careful consideration, we feel that it has merit but does not fully meet PLOS ONE’s publication criteria as it currently stands. Therefore, we invite you to submit a revised version of the manuscript that addresses the points raised during the review process.

Please, see reviewers comments to implement your manuscript.

Moreover, it is important to improve some aspects of the article. Specifically:

While the article presents a strong viewpoint, a more balanced discussion including counterarguments or limitations would strengthen its academic rigor.

We look forward to receiving your revised manuscript.

Kind regards,

Andrea Cioffi

Academic Editor

PLOS ONE

Journal Requirements:

[This research was supported by the Basic Science Research Program through the National Research Foundation of Korea (NRF) funded by the Ministry of Education (NRF-2022R1C1C1011730).].

Reviewers' comments:

Reviewer's Responses to Questions

**Comments to the Author**

1. Is the manuscript technically sound, and do the data support the conclusions?

Reviewer #1: Yes

Reviewer #2: Partly

Reviewer #3: Yes

Reviewer #4: Yes

2. Has the statistical analysis been performed appropriately and rigorously?

Reviewer #1: I Don't Know

Reviewer #2: I Don't Know

Reviewer #3: Yes

Reviewer #4: I Don't Know

3. Have the authors made all data underlying the findings in their manuscript fully available?

Reviewer #1: Yes

Reviewer #2: Yes

Reviewer #3: Yes

Reviewer #4: Yes

4. Is the manuscript presented in an intelligible fashion and written in standard English?

Reviewer #1: Yes

Reviewer #2: Yes

Reviewer #3: Yes

Reviewer #4: Yes

Reviewer #1: The article is a well-executed and timely study that makes significant contributions to understanding medical abortion discourse in South Korea and the US. However, addressing the outlined limitations and integrating more diverse data and methods could enhance the depth, applicability, and impact of the findings.

Main concerns:

1) The reliance on web-scraped news articles from specific platforms like Naver News and Google News may introduce a selection bias, potentially excluding diverse or less mainstream perspectives.

2) The dataset includes articles from 2006 to 2022, which may overlook temporal shifts in discourse, especially after pivotal legal or policy changes.

3) While the study attempts to adapt sentiment analysis tools for Korean and English texts, the effectiveness of these tools in capturing cultural nuances and contextual meanings remains unclear. Misinterpretation of sentiments could affect the validity of the results.

4) The exclusive focus on news articles neglects other influential media forms, such as social media or public opinion surveys, which might provide a broader understanding of societal attitudes.

Reviewer #2: Dear Authors,

I read with great interest your article on abortion. This study will use AI-driven text analysis (topic modeling and sentiment analysis) to examine and compare how abortion with mifepristone is discussed in South Korea and the US. It will analyze social issues, legal debates, cultural attitudes, and public sentiment surrounding medical abortion, offering insights into how the discourse varies across different legal, ethical, and cultural landscapes. My concern is that if there is no human input to choose which articles to be included in the analysis, then it may mean that you are including everything which may not reflect the actual social, cultural and legal attitudes. google and other sites are full of non-sense information. I am not sure if we can draw any valuable conclusions if we are not fully aware of what we are analysing.

Reviewer #3: General Comment

I sincerely believe this is an excellent piece of work, carefully designed, well-executed, and meaningful in its scope and contribution.

Abstract

The abstract serves as the gateway and first impression of the manuscript. I recommend that you make it more comprehensive and inclusive, offering a clear overview of the study's objectives, methods, key findings, and conclusions. This will help readers grasp the essence of your work at a glance.

Background

Well done. However, to strengthen this section, I suggest expanding on the research problem and the significance of the study. Additionally, it would be valuable to discuss the global use and prevalence of mifepristone. Including relevant data from the World Health Organization (WHO) could provide useful context and reinforce the importance of your research.

Methods

• This section would benefit from a more detailed explanation of the platform selection process for data extraction. Please clarify the steps and procedures followed, including the criteria for inclusion and exclusion of data sources.

• Furthermore, a brief but clear description of your measurement procedures would enhance the methodological transparency of the study.

Results and Discussion

You have effectively presented and interpreted your findings. The analysis is insightful and well-connected to the research questions. This section is a strength of the manuscript. Well done.

Conclusion

Since this is the final impression of your work, I suggest refining it slightly. Highlight the key findings, emphasize their wider relevance, and reinforce the significance of the study. A strong conclusion provides a sense of closure and impact for the reader.

References

Some references appear not to fully align with the journal’s formatting guidelines. Please ensure that all citations strictly follow the required style.

Reviewer #4: I would like to thank the authors for the opportunity to review this manuscript. The study presents an interesting comparative analysis of media discourse surrounding mifepristone and medical abortion in South Korea and the United States using text mining techniques.

However, there are several areas that would benefit from revision

1- In the introduction: what are the specific gaps in knowledge that this study aims to address. And why these two countries?

2- The data collection process lacks transparency regarding the search terms used for web crawling. The authors mention using Google News and Naver News but do not specify the exact search queries employed

3- The authors should provide more information about the criteria used for including or excluding news articles. Was there any screening process to ensure the articles were specifically focused on mifepristone rather than abortion more broadly?

4- The significant disparity in sample size between Korean (7,938) and US (650) articles raises concerns about comparability. The authors should address how this imbalance might affect the interpretation of results and whether any normalization techniques were applied to account for this difference.

5- Figure 1 (Pipeline of the text analysis) is referenced but not adequately described in the text.

6- grammatical errors and awkward phrasings throughout the manuscript; e.g:

- line 41: "including and is extensively prescribed" → delete "including."

- Line 68: "Comparative analyses of news content between countries can provide" → "Comparative analyses of news content can provide."

- Page 1, Abstract: "distinct pattern" should be "distinct patterns".

7- The abbreviations should be defined at first use and then used consistently throughout the manuscript.

**Do you want your identity to be public for this peer review?** For information about this choice, including consent withdrawal, please see our Privacy Policy

Reviewer #1: No

Reviewer #2: **Yes:** Nourah Hasan Al Qahtani

Reviewer #3: No

Reviewer #4: No

---

## [Author Response · Author response to Decision Letter 1]

15 Jul 2025

Reviewer 1.

The article is a well-executed and timely study that makes significant contributions to understanding medical abortion discourse in South Korea and the US. However, addressing the outlined limitations and integrating more diverse data and methods could enhance the depth, applicability, and impact of the findings.

1) The reliance on web-scraped news articles from specific platforms like Naver News and Google News may introduce a selection bias, potentially excluding diverse or less mainstream perspectives.

Response:

Thank you very much for your valuable comment.

As you pointed out, our study primarily utilized web-scraped news articles from Naver News and Google News. However, we selected these platforms due to their prominence and representativeness in their respective countries. Specifically, Naver News is the most widely used news aggregator in South Korea, extensively covering diverse news outlets and thus effectively capturing mainstream discourse. Similarly, Google News is the leading news aggregation platform in the United States, encompassing a broad spectrum of major media sources and ensuring the reflection of dominant public narratives.

To clarify and emphasize the representativeness and rationale for choosing these platforms, we have added the following statement in the Methods section of the revised manuscript:

To ensure the reliability and relevance of the collected articles, only articles from registered news media sources indexed by Google News and Naver News were included. Naver News was considered for South Korea owing to its dominant position in aggregating nearly all major domestic news outlets [22], and Google News was selected for the US because of its extensive and representative aggregation of leading American news sources [23]. Both platforms have news policies [24–25] to ensure that the content they provide comes from media outlets that adhere to best practices.

2) The dataset includes articles from 2006 to 2022, which may overlook temporal shifts in discourse, especially after pivotal legal or policy changes.

Response:

We are grateful for the valuable feedback. The study period (2006–2022) was strategically selected to capture critical legal and policy shifts related to medical abortion. We have clarified the rationale for this choice in the discussion section of the revised manuscript as the following:

Additionally, our analysis spans news articles between 2006 and 2022, potentially introducing a time-trend bias, as public discourse on media perspectives possibly evolved over this period. However, the study period (2006–2022) was specifically selected to capture critical legal and policy shifts related to abortion, enabling a thorough analysis of how media discourse adapted to these temporal shifts. This period saw global and regional regulatory milestones regarding mifepristone. In February 2006, the Federal Parliament of Australia overturned the Harradine Amendment—a legislative restriction that had previously limited the availability of mifepristone [41]. Moreover, in South Korea, the selected timeframe captured the 2017 Blue House petition demanding reconsideration of abortion regulations, the Constitutional Court’s landmark 2019 ruling deeming the existing abortion ban unconstitutional, and the subsequent abolishment of abortion restrictions in January 2021. The observed increase in discourse following Hyundai Pharmaceutical’s initial submission for mifepristone approval in 2021 further validates that the analyzed period encompassed key events that likely influenced media narratives. Thus, the extended analysis period adequately addressed potential concerns about overlooking temporal shifts in media discourse, illustrating how public narratives evolved alongside these landmark events. Future research could address these limitations by incorporating more news sources, employing dynamic topic modeling to capture temporal trends, and integrating qualitative methods.

3) While the study attempts to adapt sentiment analysis tools for Korean and English texts, the effectiveness of these tools in capturing cultural nuances and contextual meanings remains unclear. Misinterpretation of sentiments could affect the validity of the results.

Response:

We sincerely appreciate the reviewer’s insightful comment regarding potential limitations associated with the sentiment analysis tools used for Korean and English texts. As pointed out, there is a possibility of misinterpretation due to cultural nuances and contextual differences across languages. We fully acknowledge this limitation, and we have explicitly addressed this point in the Discussion section of our revised manuscript. Specifically, we added references that demonstrate the validity and common usage of these sentiment analysis tools in cross-cultural text mining studies, while simultaneously emphasizing the importance of cautious interpretation when applying lexicon-based approaches to culturally distinct datasets.

We acknowledge a potential limitation regarding the effectiveness of lexicon-based sentiment analysis tools in accurately capturing the cultural nuances and contextual meanings inherent in Korean and English texts. While these tools have been validated and commonly used across diverse linguistic and cultural contexts in previous research [38–40], there remains a possibility of misinterpretation when applied to culturally distinct datasets. To mitigate it, the researchers involved in this study who are proficient in both languages cross-validated and any disagreements regarding contextual interpretation beyond automated sentiment analysis outputs were resolved through discussion. Nevertheless, our sentiment analysis results should be interpreted with caution, and future research should consider additional qualitative analyses or culturally customized lexicons to better account for language-specific subtleties.

4) The exclusive focus on news articles neglects other influential media forms, such as social media or public opinion surveys, which might provide a broader understanding of societal attitudes.

Response:

Thank you very much for highlighting this important point regarding our exclusive reliance on news articles. We fully acknowledge that incorporating social media or public opinion survey data could offer additional valuable perspectives on societal attitudes toward medical abortion. However, our research specifically aimed to capture institutional media representations of regulatory, societal, and policy discussions rather than individual, potentially biased viewpoints that frequently appear in social media. We have clarified this focus in the revised discussion section, emphasizing that social media data, while valuable, could introduce individual-level biases and distortions. Nevertheless, we suggest that future studies incorporate social media or public opinion data as complementary resources to expand and enrich the understanding of societal attitudes towards medical abortion.

This study had some limitations. While various big data open-sources are available for text mining, our analysis relied on web-crawled news articles from specific platforms and periods, which may not capture the full spectrum of media discourse on mifepristone. This study analyzed news articles to accurately capture formal media representations of regulatory, societal, and policy issues regarding medical abortion. Although leveraging social media content for the analysis could provide additional insights, individual opinions expressed through social media platforms usually introduce significant biases and distortions, precluding the accurate assessment of institutional discourse.

Reviewer #2:

I read with great interest your article on abortion. This study will use AI-driven text analysis (topic modeling and sentiment analysis) to examine and compare how abortion with mifepristone is discussed in South Korea and the US. It will analyze social issues, legal debates, cultural attitudes, and public sentiment surrounding medical abortion, offering insights into how the discourse varies across different legal, ethical, and cultural landscapes. My concern is that if there is no human input to choose which articles to be included in the analysis, then it may mean that you are including everything which may not reflect the actual social, cultural and legal attitudes. google and other sites are full of non-sense information. I am not sure if we can draw any valuable conclusions if we are not fully aware of what we are analysing.

Response:

We appreciate the reviewer’s important comment. The description regarding sorting out irrelevant or duplicate articles was inadvertently omitted in the initial manuscript. We have now clearly described that only articles from registered and reputable news media indexed by Google and Naver were utilized, and irrelevant or duplicated articles were systematically excluded through manual review and automated deduplication processes. This clarification has been added to the Methods section in the revised manuscript. We sincerely appreciate your insightful suggestion.

To ensure the reliability and relevance of the collected articles, only articles from registered news media sources indexed by Google News and Naver News were included. Naver News was considered for South Korea owing to its dominant position in aggregating nearly all major domestic news outlets [22], and Google News was selected for the US because of its extensive and representative aggregation of leading American news sources [23]. Both platforms have news policies [24–25] to ensure that the content they provide comes from media outlets that adhere to best practices. Duplicate articles and those irrelevant to the study’s context were identified and excluded through manual sorting and automated deduplication processes.

Reviewer #3:

I sincerely believe this is an excellent piece of work, carefully designed, well-executed, and meaningful in its scope and contribution.

Abstract

The abstract serves as the gateway and first impression of the manuscript. I recommend that you make it more comprehensive and inclusive, offering a clear overview of the study's objectives, methods, key findings, and conclusions. This will help readers grasp the essence of your work at a glance.

Response:

Thank you very much for your valuable feedback on our abstract.

Following your suggestion, we have revised the abstract to clearly and comprehensively present the study's objective, methods, key findings, and conclusions. We believe this updated version effectively summarizes the essence of our research and provides readers with a clearer overview.

Background

Well done. However, to strengthen this section, I suggest expanding on the research problem and the significance of the study. Additionally, it would be valuable to discuss the global use and prevalence of mifepristone. Including relevant data from the World Health Organization (WHO) could provide useful context and reinforce the importance of your research.

Response:

Thank you for your valuable comments on strengthening our introduction.

As suggested, we have expanded our introduction to clearly articulate the research problem and significance of this study. Additionally, we included relevant data from the World Health Organization (WHO) to provide global context regarding the prevalence and recommended use of mifepristone, thereby reinforcing the importance and relevance of our research.

The revised introduction section is presented below.

Mifepristone, a common oral abortifacient, is legally available in numerous countries, and the World Health Organization (WHO) has endorsed the combined use of mifepristone and misoprostol as safe, effective, and highly acceptable regimen for early abortion—defined as abortion during the first trimester [3]. In 2019, the WHO moved these medications from the “complementary” to the “core” list of its Essential Medicines List, given their critical role in global reproductive health [4].

Following these recommendations, the global prevalence of medical abortion has increased, particularly in high-income countries [5]. By 2017, medical abortion comprised more than one-half of all abortion provisions in at least 24 high-income nations, with high proportions in Sweden, England, and the United States (US). In the US in 2023, approximately 63% of all abortions were medication-based abortions [5]. Nevertheless, regulatory approval and access to mifepristone vary among countries; for instance, despite its use in the US [5–7], it remains unapproved in South Korean, resulting in limited availability of medical abortion [8].

Methods

• This section would benefit from a more detailed explanation of the platform selection process for data extraction. Please clarify the steps and procedures followed, including the criteria for inclusion and exclusion of data sources.

Response:

Thank you for this insightful suggestion.

We have now clarified and expanded upon the platform selection process in the Methods section, specifying detailed criteria used for including or excluding data sources. This revision highlights the rationale behind the selection of Naver News and Google News platforms, including their representativeness and comprehensiveness, and explicitly describes the inclusion/exclusion criteria employed during article collection.

The revised Methods section is provided below.

To ensure the reliability and relevance of the collected articles, only articles from registered news media sources indexed by Google News and Naver News were included. Naver News was considered for South Korea owing to its dominant position in aggregating nearly all major domestic news outlets [22], and Google News was selected for the US because of its extensive and representative aggregation of leading American news sources [23]. Both platforms have news policies [24–25] to ensure that the content they provide comes from media outlets that adhere to best practices. Duplicate articles and those irrelevant to the study’s context were identified and excluded through manual sorting and automated deduplication processes.

Methods

• Furthermore, a brief but clear description of your measurement procedures would enhance the methodological transparency of the study.

Response:

Thank you for your suggestion. We fully agree that including a brief yet clear explanation of the measurement procedures would enhance the methodological transparency of our study. The revised Methods sections reflecting your comment is provided below.

The study comprised three stages: (1) data collection and preprocessing, (2) topic modeling using Latent Dirichlet Allocation (LDA), and (3) sentiment analysis using lexicon-based methods. The outline of the text analysis pipeline is described in Fig 1.

Conclusion

Since this is the final impression of your work, I suggest refining it slightly. Highlight the key findings, emphasize their wider relevance, and reinforce the significance of the study. A strong conclusion provides a sense of closure and impact for the reader.

Response:

We have refined the conclusion to clearly highlight the key findings, emphasize their broader implications, and reinforce the importance of our research. The revised conclusion is presented below.

This study identifies differences in media discourse on medical abortion using mifepristone between South Korea and the US, reflecting differences in the number of articles, key topics, and distinctive patterns of sentiments in each country. The topic modelling results showed that the Korean news focuses on regulatory challenges, safety concerns, and social perceptions, whereas the US coverage underscores clinical aspects and access-related debates. Our findings reveal differing public concerns and interests between the two countries, which may assist policymakers and healthcare providers in developing strategies to navigate complex societal dynamics surrounding medical abortion.

References

Some references appear not to fully align with the journal’s formatting guidelines. Please ensure that all citations strictly follow the required style.

Response:

Thank you for highlighting the formatting issue. We carefully reviewed and rev

---

## [Decision Letter · Decision Letter 1]

17 Sep 2025

Dear Dr. Song,

Thank you for submitting your manuscript to PLOS ONE. After careful consideration, we feel that it has merit but does not fully meet PLOS ONE’s publication criteria as it currently stands. Therefore, we invite you to submit a revised version of the manuscript that addresses the points raised during the review process.

We note that some of the comments of the reviewers refer to specific articles for you to cite. Please note that it is not mandatory that you cite these specific articles and you are welcome to seek alternatives manuscripts in the literature that are relevant to your manuscript’s content.

We look forward to receiving your revised manuscript.

Kind regards,

Andrea Cioffi

Academic Editor

PLOS ONE

Journal Requirements:

Reviewers' comments:

Reviewer's Responses to Questions

**Comments to the Author**

Reviewer #1: All comments have been addressed

Reviewer #3: All comments have been addressed

Reviewer #5: (No Response)

2. Is the manuscript technically sound, and do the data support the conclusions?

Reviewer #1: Yes

Reviewer #3: Yes

Reviewer #5: Partly

3. Has the statistical analysis been performed appropriately and rigorously?

Reviewer #1: Yes

Reviewer #3: Yes

Reviewer #5: I Don't Know

4. Have the authors made all data underlying the findings in their manuscript fully available?

Reviewer #1: Yes

Reviewer #3: Yes

Reviewer #5: Yes

5. Is the manuscript presented in an intelligible fashion and written in standard English?

Reviewer #1: Yes

Reviewer #3: Yes

Reviewer #5: Yes

Reviewer #1: The revised manuscript effectively addresses the prior concerns and now presents a well-structured, ethically sensitive, and scientifically sound analysis with just minor refinements as noted below:

1.Ethical Clarity: You mention ethics approval and waiver of consent, but do not explicitly state how sensitive abortion data were handled (e.g., anonymisation, database separation, encryption).

2.Cultural Context (Discussion section): A brief acknowledgement that the findings may not be generalizable to settings where abortion is legally restricted or stigmatised would enhance the discussion’s depth.

3.Clinical Significance: The adjusted odds ratios for biochemical pregnancy and clinical pregnancy are slightly reduced but not statistically significant. Consider adding a sentence to discuss whether this might still be clinically meaningful or merits further investigation in a larger cohort.

Reviewer #3: All comments and suggestions have been adequately addressed. The authors have made significant improvements, and I believe the work is now ready for acceptance.

Reviewer #5: I would like to sincerely thank the Editor for the opportunity to review a manuscript submitted to such a highly esteemed journal. It is an honour for me.

It has been a pleasure to read this article, which I found to be well-structured and highly original in both its methodological approach and thematic focus. Nevertheless, I believe that some revisions are necessary before the manuscript can be recommended for publication. I hope that the following suggestions will support the authors in strengthening their work and guiding their revisions.

- The Introduction is solid, well-documented, and clearly structured, with a strong rationale for the study. However, improvements in language style, clarity, and conciseness would enhance the manuscript’s readability and impact, particularly for an international audience. Additionally, several grammatical issues require attention before publication. For instance, the phrase “in South Korean” should be corrected to “in South Korea,” and expressions such as “concerns persist safety of mifepristone” are grammatically problematic. Terms such as “medial abortion” also need revision for accuracy. Moreover, the legal and regulatory references appear somewhat imbalanced: while the US context is discussed in detail, the Korean legal framework receives more limited attention. A more thorough and globally contextualized background would be beneficial. In particular, integrating European examples would offer a more comprehensive overview of international approaches to medical abortion—especially since the conclusions highlight global policy implications. To that end, I recommend including references such as:

https://pubmed.ncbi.nlm.nih.gov/34897104/

https://pubmed.ncbi.nlm.nih.gov/12831608/

https://pubmed.ncbi.nlm.nih.gov/12137129/

These sources can enrich the comparative dimension of the analysis and support a more robust global framing.

- The Methods section is clearly structured and methodologically sound. However, unless I have overlooked it, the initial number of articles retrieved per country (prior to filtering or selection) is not explicitly stated—this detail would improve transparency. Additionally, I would suggest adding a brief note regarding the reproducibility of the research process (e.g., availability of code, corpora, or full protocol), which is particularly important in computational text analysis.

- The Results are presented in a clear and orderly manner, and the comparative analysis between South Korea and the US is effective. To further support interpretation, I suggest including a short summary section that explicitly highlights the key differences observed between the two countries in terms of themes and sentiments.

- In the Discussion, two aspects warrant careful attention:

Avoid implying causal relationships that are not empirically grounded. For example, the assumption that illegal online trade results directly from regulatory restrictions should be carefully qualified as a possible causal link or interpretative hypothesis, unless further evidence is provided.

The influence of media narratives on public opinion is mentioned but not sufficiently explored. A brief reflection on how and why media framing can shape public understanding—supported by relevant literature—would add theoretical depth and policy relevance to the discussion.

- The Conclusion successfully summarizes the findings but remains rather general. It would benefit from being more precise and action-oriented, particularly by offering concrete recommendations. For example, the authors could suggest how media coverage might adopt a more balanced and informative narrative regarding medical abortion, or how policymakers might engage with media discourses more strategically.

**Do you want your identity to be public for this peer review?** For information about this choice, including consent withdrawal, please see our Privacy Policy

Reviewer #1: **Yes:** JACKLINE AKELLO

Reviewer #3: No

Reviewer #5: No

---

## [Author Response · Author response to Decision Letter 2]

30 Oct 2025

Reviewer 1.

The revised manuscript effectively addresses the prior concerns and now presents a well-structured, ethically sensitive, and scientifically sound analysis with just minor refinements as noted below:

1) Ethical Clarity: You mention ethics approval and waiver of consent, but do not explicitly state how sensitive abortion data were handled (e.g., anonymisation, database separation, encryption).

Response:

Thank you very much for your comments with ethical clarity.

We agree with the importance of clarifying the potential sensitivity regarding the data. However, our study did not involve any human subjects or personally identifiable information. The dataset we used consisted exclusively of publicly available news articles crawled from online news platforms. No individual-level or sensitive health data were included, and all analyses were performed on anonymized textual data in aggregate form.

We have added a clarifying statement in the Methods section as the following:

This study was a retrospective comparative analysis of published news articles using a text-mining approach. As the study was conducted using publicly available online news articles without any non-identifiable data and that did not involve human participants, no sensitive or confidential data were included in the dataset, and all analyses were conducted on aggregated, anonymized text corpora. Therefore, it was exempted from ethical review by the Institutional Review Board (IRB) of Daegu Catholic University (IRB No. CUIRB-2022-E004).

2) Cultural Context (Discussion section): A brief acknowledgement that the findings may not be generalizable to settings where abortion is legally restricted or stigmatised would enhance the discussion’s depth.

Response:

We appreciate the suggestion. We agree that the generalizability of our findings may be limited based on legal restrictions or social stigmas. While we already have noted the need for cultural sensitivity in interpreting sentiment analysis results, we further expanded this part in Discussion section to explicitly acknowledge that media discourse and public sentiment toward medical abortion can be shaped by distinct cultural, legal, and ethical environments as suggested.

It should also be noted that the generalizability of these findings may be limited as media narratives and public sentiment towards medical abortion are influenced by the unique cultural, legal, and ethical environment.

3) Clinical Significance: The adjusted odds ratios for biochemical pregnancy and clinical pregnancy are slightly reduced but not statistically significant. Consider adding a sentence to discuss whether this might still be clinically meaningful or merits further investigation in a larger cohort.

Response:

We appreciate the reviewer’s thorough evaluation. We revised the discussion section as follows:

Sentiment analysis on news coverage surrounding mifepristone in each country provided insights into the public perceptions. The US articles displayed a polarized pattern, with strongly positive sentiments associated with the term 'treatment' and negative sentiments linked to 'restrictions'. Other terms, such as 'women,' remained relatively neutral in the US dataset. In contrast, Korean articles demonstrated more varied sentiment distributions. Terms such as 'legalize', 'petition', and 'abolish' conveyed negative sentiments, reflecting both the restrictive environment and moral concerns surrounding abortion, while words like 'overseas,' 'treatment,' and 'safety' elicited positive sentiments. Notably, the term 'abortion' was associated with near-neutral sentiment in Korean texts, potentially indicating changing public attitudes and reflecting ongoing challenges in the societal acceptance of medical abortion in South Korea [13,32]. The neutrality associated with the word 'permit' and slight negativity with terms such as 'legalize' and 'sale' in Korean articles highlight the underlying ambivalence and moral unease within public discourse. Such sentiment patterns suggest persistent societal resistance and caution toward the acceptance of medical abortion, reflecting broader apprehensions about the moral and health implications of expanding access to abortifacient drugs. These findings underscore the need for prudent regulatory approaches that prevent the normalization of medically and morally contentious practices. This contrasts with countries such as the US and Canada, where extensive evidence has contributed to widespread regulatory acceptance and clinical recognition of the effectiveness of mifepristone [36,37]. Furthermore, given the marginally neutral but somewhat negative connotation of the term ‘labor-inducing agent’, further large-scale cohort studies are warranted in countries where mifepristone is used, to assess its long-term safety in women. These findings emphasize the need for careful, value-sensitive and evidence-based policies that respect prevailing ethical perspectives while addressing emerging needs for safe clinical practices [38].

Reviewer #5:

I would like to sincerely thank the Editor for the opportunity to review a manuscript submitted to such a highly esteemed journal. It is an honour for me. It has been a pleasure to read this article, which I found to be well-structured and highly original in both its methodological approach and thematic focus. Nevertheless, I believe that some revisions are necessary before the manuscript can be recommended for publication. I hope that the following suggestions will support the authors in strengthening their work and guiding their revisions.

4) The Introduction is solid, well-documented, and clearly structured, with a strong rationale for the study. However, improvements in language style, clarity, and conciseness would enhance the manuscript’s readability and impact, particularly for an international audience. Additionally, several grammatical issues require attention before publication. For instance, the phrase “in South Korean” should be corrected to “in South Korea,” and expressions such as “concerns persist safety of mifepristone” are grammatically problematic. Terms such as “medial abortion” also need revision for accuracy. Moreover, the legal and regulatory references appear somewhat imbalanced: while the US context is discussed in detail, the Korean legal framework receives more limited attention. A more thorough and globally contextualized background would be beneficial. In particular, integrating European examples would offer a more comprehensive overview of international approaches to medical abortion—especially since the conclusions highlight global policy implications. To that end, I recommend including references such as: https://pubmed.ncbi.nlm.nih.gov/34897104/
https://pubmed.ncbi.nlm.nih.gov/12831608/
https://pubmed.ncbi.nlm.nih.gov/12137129/ These sources can enrich the comparative dimension of the analysis and support a more robust global framing.

Response:

We appreciate deeply your valuable comments and constructive suggestions. We have thoroughly revised the introduction for improved clarity, language, and conciseness, correcting all grammatical issues. Additionally, we have revised the Introduction overall to ensure a more balanced discussion of both countries. To strengthen the global perspective, relevant European examples were incorporated with proper references cited in both Introduction section as follows.

Abortion remains a socially divisive issue, balancing women’s reproductive rights against moral and ethical opposition [1]. Access to abortion services varies globally depending on legislation, social values, and healthcare policies. Medical abortion using mifepristone has raised ongoing debates regarding its moral legitimacy and potential health risks, although often described as a less invasive alternative to surgery [2]. Approved by the United States (US) Food and Drug Administration (FDA) in 2000 after several decades of use in European countries such as France, the United Kingdom. Italy and Sweden, mifepristone is now available in numerous countries, following the World Health Organization’s (WHO) endorsement of its use in combination with misoprostol as an early abortion regimen [3–6]. Prevalence of medical abortion has increased worldwide, comprising more than half of all abortions in high-income countries. In the US, despite the FDA’s regulatory oversight and the adoption of Risk Evaluation and Mitigation Strategy, reports of adverse events such as severe bleeding, ectopic pregnancy, and systemic infections persist [7–11]. In contrast, South Korea has yet to approve mifepristone despite the 2019 Constitutional Court ruling decriminalizing abortion up to 14 weeks of gestation [12,13]. Cultural stigma, conservative values, and limited public discourse on reproductive ethics continue to restrict access, underscoring the need for cautious, evidence-based policymaking than balances women’s health with broader moral and societal consideration [13].

Media discourse plays a pivotal role in shaping public attitudes and policy debates on contested topics such as abortion [14,15]. Coverage of medical abortion and mifepristone reflects prevailing societal attitudes, cultural beliefs, and political contexts. Comparative analyses of news content across countries can illuminate how differing moral climates and policy priorities generate divergent narratives [16,17]. Such analysis can elucidate underlying factors that contribute to disparate approaches towards medical abortion, thereby supporting the evidence-based communication and policy strategies. In the digital era, computational text-mining techniques such as topic modelling and sentiment analysis enable systemic exploration of dominant themes, sentiment trends, and linguistic patterns in media representations of health issues [18]. However, limited research has compared media discourse on medical abortion across countries with contrasting regulatory and cultural contexts, notably South Korea and the US. Understanding these differences is crucial to evaluating how societal values and policy frameworks shape media narratives, thereby informing culturally sensitive and empirically grounded reproductive health policymaking.

Therefore, this study aims to comprehensively compare news media coverage in South Korea and the US through topic modeling and sentiment analysis, identifying predominant themes and sentiment patterns surrounding medical abortion with mifepristone. By clarifying how distinct national contexts influence media narratives and public perceptions, this research seeks to provide critical insights to guide policymakers, healthcare providers, and advocacy groups in developing culturally informed reproductive health policies.

5) The Methods section is clearly structured and methodologically sound. However, unless I have overlooked it, the initial number of articles retrieved per country (prior to filtering or selection) is not explicitly stated—this detail would improve transparency. Additionally, I would suggest adding a brief note regarding the reproducibility of the research process (e.g., availability of code, corpora, or full protocol), which is particularly important in computational text analysis.

Response:

We sincerely appreciate your valuable comments regarding methodological transparency and reproducibility. As noted in the revised manuscript, we included only news articles that contained mifepristone-related keywords in either the title or the main text. Duplicate articles were automatically removed using a pre-specified script, after which all remaining articles were manually reviewed; no additional exclusions were necessary. Therefore, the 7,938 Korean and 650 U.S. news articles reported in the manuscript represent the final datasets used for the analysis.

Regarding reproducibility, we acknowledge that the exact number of articles or corpora may vary in the future if certain news articles are deleted by publishers or if platform crawling policies are modified. Nevertheless, to enhance research transparency, we have added a statement at the end of the manuscript indicating that the analysis code is available from the authors upon reasonable request.

For greater clarity, the manuscript has also been revised as follows:

The search included news articles that contained the keywords “mifepristone,” “misoprostol,” “Mifegyne,” or “RU-486” in either the title or the main text. On Korean platforms, these keywords were searched in Korean, while English terms were used for the U.S. platforms……. There were no irrelevant articles identified after manual sorting.

Data Availability Statement

The datasets analyzed in this study were derived from publicly available online news articles retrieved from Google News and Naver News between January 2006 and December 2022. Both the processed datasets and the analysis code used in this study are available from the corresponding author upon reasonable request. However, the exact number of accessible articles or the composition of the corpora may vary in the future depending on changes in news outlet availability or platform crawling policies.

6) The Results are presented in a clear and orderly manner, and the comparative analysis between South Korea and the US is effective. To further support interpretation, I suggest including a short summary section that explicitly highlights the key differences observed between the two countries in terms of themes and sentiments.

Response:

Thank you for this helpful suggestion. We agree that adding a concise summary of the key differences would enhance readers’ understanding of the results. Since the Results section is organized by four analytical components with specific subheadings, adding an additional paragraph at the end could disrupt the structural flow and cause redundancy. Therefore, we have incorporated a clear summary paragraph at the beginning of the Discussion section as follows.

This study examines the social context of mifepristone in South Korea and the US through an analysis of news media discourse using medical informatics methodology. Leveraging analytical methods including text mining, topic modeling, and sentiment analysis, we identified prominent topics, underlying sentiments, and word associations that illustrate public attitudes and policy discourse regarding medical abortion within regulatory and healthcare contexts. The findings revealed media coverage in South Korea primarily emphasized the regulatory approval process, safety concerns, and social controversies. In contrast, the US media discourse was characterized by legal debates, clinical applications, and accessibility, with more polarized sentiment patterns.

7) In the Discussion, two aspects warrant careful attention: Avoid implying causal relationships that are not empirically grounded. For example, the assumption that illegal online trade results directly from regulatory restrictions should be carefully qualified as a possible causal link or interpretative hypothesis, unless further evidence is provided. The influence of media narratives on public opinion is mentioned but not sufficiently explored. A brief reflection on how and why media framing can shape public understanding—supported by relevant literature—would add theoretical depth and policy relevance to the discussion.

Response:

We sincerely appreciate this insightful comment. In response, we have carefully reviewed and revised the Discussion section, integrating relevant literatures as follows.

Furthermore, our study noted the illegal online trade of mifepristone frequently reported in Korean news coverage. Such reporting may suggest possible unintended associations between stringent regulatory controls and unsafe practices and misuse, raising public health and safety concerns [34]. These observations underscore the importance of developing balanced, evidence-based regulatory frameworks prioritizing public health and moral responsibility. Despite the role of legal changes in abortion debates, the word ‘law’ did not appear among the terms with high TF-IDF values in Korean articles. This omission may indicate a strong media focus on practical considerations related to the introduction of medical abortion rather than on lega

---

## [Decision Letter · Decision Letter 2]

11 Nov 2025

Dear Dr. Song,

Thank you for submitting your manuscript to PLOS ONE. After careful consideration, we feel that it has merit but does not fully meet PLOS ONE’s publication criteria as it currently stands. Therefore, we invite you to submit a revised version of the manuscript that addresses the points raised during the review process.

We look forward to receiving your revised manuscript.

Kind regards,

Andrea Cioffi

Academic Editor

PLOS ONE

Journal Requirements:

Additional Editor Comments:

Please note that it is very important to submit a manuscript with all the changes highlighted.

Reviewers' comments:

Reviewer's Responses to Questions

**Comments to the Author**

Reviewer #5: (No Response)

2. Is the manuscript technically sound, and do the data support the conclusions?

Reviewer #5: Partly

3. Has the statistical analysis been performed appropriately and rigorously?

Reviewer #5: I Don't Know

4. Have the authors made all data underlying the findings in their manuscript fully available?

Reviewer #5: Yes

5. Is the manuscript presented in an intelligible fashion and written in standard English?

Reviewer #5: Yes

Reviewer #5: I would like to thank the Editor once again for the opportunity to review this manuscript. However, I am unable to clearly identify the revisions made in response to my previous comments. Although the authors provide a point-by-point reply, these responses do not always appear to correspond in a clear and consistent manner to changes in the main text.

I would therefore kindly ask the authors to submit a newly revised version of the manuscript, in clean format, in which only the additions, modifications, and new references introduced in response to the previous review are clearly highlighted—for example, marked in red—so as to allow a more accurate assessment of the adequacy of the revisions undertaken.

**Do you want your identity to be public for this peer review?** For information about this choice, including consent withdrawal, please see our Privacy Policy

Reviewer #5: No

---

## [Author Response · Author response to Decision Letter 3]

12 Dec 2025

Response to reviewers

Andrea Cioffi

Academic Editor, Plos One

Dear Cioffi,

We resubmit for publication the revised version of PONE-D-25-00622R2 “Comparative Analysis of Social Issues Towards Medical Abortion Using Mifepristone in South Korea and the US: Topic Modelling and Sentiment Analysis.” We thank you and the reviewers for their critical assessment and thoughtful suggestions of our work. We sincerely apologize to Reviewer #5 for the inconvenience caused by the lack of clarity in our previous revision. We fully understand that the difficulty in identifying the specific changes may have created unnecessary burden during the review process.

We hope that the revised manuscript is now suitable for publication in PLOS ONE.

Sincierely,

Yun-Kyoung Song, Ph.D. (Corresponding author)

College of Pharmacy, The Catholic University of Korea-Sungsim Campus, Bucheon, Gyeonggido, 14662, Republic of Korea

E-mail address: yksong@catholic.ac.kr

Tel: +82-2-2164-5515, +82-10-5099-9741

Reviewer #5:

I am unable to clearly identify the revisions made in response to my previous comments. Although the authors provided a point-by-point reply, these responses do not always correspond in a clear and consistent manner to changes in the main text.

I therefore kindly ask the authors to submit a newly revised version of the manuscript, in clean format, in which only the additions, modifications, and new references introduced in response to the previous review are clearly highlighted-for example, marked in red-so as to allow a more accurate assessment of the adequacy of the revisions undertaken.

Response:

We sincerely thank Reviewer #5 for the careful reading of our manuscript and for pointing out the difficulty in identifying the exact revisions made in the previous round. We deeply appreciate the reviewer’s patience and the opportunity to further clarify and improve the transparency of our revision process.

In this resubmission, we have taken several steps to ensure that all revisions are fully traceable, clearly highlighted, and directly matched to each comment. In addition, as we substantially revised the Introduction to improve clarity and coherence, only newly added text has been highlighted in red.

Reviewer’s Comment: The Introduction is solid, well-documented, and clearly structured, with a strong rationale for the study. However, improvements in language style, clarity, and conciseness would enhance the manuscript’s readability and impact, particularly for an international audience. Additionally, several grammatical issues require attention before publication. For instance, the phrase “in South Korean” should be corrected to “in South Korea,” and expressions such as “concerns persist safety of mifepristone” are grammatically problematic. Terms such as “medial abortion” also need revision for accuracy.

Response: We have extensively revised the overall narrative and structure of the introduction to improve clarity, accuracy, and coherence in response to the reviewer’s comments.

[Location: Introduction section]

(2nd Revision) for instance, despite its use in the US [5–7], it remains unapproved in South Korean, resulting in limited availability of medical abortion [8].

-> (3rd Revision) In contrast, South Korea has yet to approve mifepristone despite the 2019 Constitutional Court ruling decriminalizing abortion up to 14 weeks of gestation [12,13].

(2nd Revision) In the US, the Food and Drug Administration (FDA) has taken a proactive, two-track approach to increasing access to medial abortion by approving a generic version of mifepristone, -> (deleted)

(2nd Revision) Even with these precautions, concerns persist safety of mifepristone, particularly risks of severe bleeding, ectopic pregnancy, and systemic infections [10,11]. -> (3rd Revision) In the US, despite the FDA’s regulatory oversight and the adoption of Risk Evaluation and Mitigation Strategy, reports of adverse events such as severe bleeding, ectopic pregnancy, and systemic infections persist [7–11].

Reviewer’s Comment Moreover, the legal and regulatory references appear somewhat imbalanced: while the US context is discussed in detail, the Korean legal framework receives more limited attention. A more thorough and globally contextualized background would be beneficial. In particular, integrating European examples would offer a more comprehensive overview of international approaches to medical abortion—especially since the conclusions highlight global policy implications. To that end, I recommend including references such as:

https://pubmed.ncbi.nlm.nih.gov/34897104/

https://pubmed.ncbi.nlm.nih.gov/12831608/

https://pubmed.ncbi.nlm.nih.gov/12137129/

These sources can enrich the comparative dimension of the analysis and support a more robust global framing.

Response: We have further elaborated the introduction section to provide a more thorough and globally contextualized background, including European examples, to offer a more comprehensive overview of international approaches to medical abortion as follows.

[Location: Introduction section]

(2nd Revision) Mifepristone, a common oral abortifacient, is legally available in numerous countries, and the World Health Organization (WHO) has endorsed the combined use of mifepristone and misoprostol as safe, effective, and highly acceptable regimen for early abortion—defined as abortion during the first trimester [3]. In 2019, the WHO moved these medications from the “complementary” to the “core” list of its Essential Medicines List, given their critical role in global reproductive health [4].

Following these recommendations, the global prevalence of medical abortion has increased, particularly in high-income countries [5]. By 2017, medical abortion comprised more than one-half of all abortion provisions in at least 24 high-income nations, with high proportions in Sweden, England, and the United States (US). In the US in 2023, approximately 63% of all abortions were medication-based abortions [5]. Nevertheless, regulatory approval and access to mifepristone vary among countries; for instance, despite its use in the US [5–7], it remains unapproved in South Korean, resulting in limited availability of medical abortion [8].

In the US, the Food and Drug Administration (FDA) has taken a proactive, two-track approach to increasing access to medial abortion by approving a generic version of mifepristone, while simultaneously addressing safety concerns through a Risk Evaluation and Mitigation Strategy (REMS) [9]. The REMS program includes healthcare provider certification, comprehensive patient education, and post-marketing surveillance monitoring to address potential risks associated with mifepristone use. Even with these precautions, concerns persist safety of mifepristone, particularly risks of severe bleeding, ectopic pregnancy, and systemic infections [10,11].

Recently, public advocacy has increased to ensure mifepristone’s market availability in the US [12]. Meanwhile, in South Korea, mifepristone remains unapproved and inaccessible via legal channels [13] for reasons that include cultural stigma associated with abortion, limited public discourse surrounding reproductive rights, and the prevailing influence of conservative societal values [14]. However, these restrictions have raised debates about women’s health, underscoring the need for evidence-based policies prioritizing reproductive autonomy [15].

-> (3rd Revision) Approved by the United States (US) Food and Drug Administration (FDA) in 2000 after several decades of use in European countries such as France, the United Kingdom. Italy and Sweden, mifepristone is now available in numerous countries, following the World Health Organization’s (WHO) endorsement of its use in combination with misoprostol as an early abortion regimen [3�6]. Prevalence of medical abortion has increased worldwide, comprising more than half of all abortions in high-income countries. In the US, despite the FDA’s regulatory oversight and the adoption of Risk Evaluation and Mitigation Strategy, reports of adverse events such as severe bleeding, ectopic pregnancy, and systemic infections persist [7–11]. In contrast, South Korea has yet to approve mifepristone despite the 2019 Constitutional Court ruling decriminalizing abortion up to 14 weeks of gestation [12,13]. Cultural stigma, conservative values, and limited public discourse on reproductive ethics continue to restrict access, underscoring the need for cautious, evidence-based policymaking than balances women’s health with broader moral and societal consideration [13].

We revised the introduction section using the references recommended by the reviewer, as follows.

[Location: References section]

(3rd Revision) 3. Jones RK, Henshaw SK. Mifepristone for early medical abortion: experiences in France, Great Britain and Sweden. Perspect Sex Reprod Health. 2002;34(3):154–61.

4. Arisi E. Changing attitudes towards abortion in Europe. Eur J Contracept Reprod Health Care. 2003;8(2):109-21.

Reviewer’s Comment The initial number of articles retrieved per country (prior to filtering or selection) is not explicitly stated—this detail would improve transparency. Additionally, I would suggest adding a brief note regarding the reproducibility of the research process (e.g., availability of code, corpora, or full protocol), which is particularly important in computational text analysis.

[Location: Method section]

(2nd Revision) The search words included “Mifepristone,” “Misoprostol,” “Mifegyne,” and “RU-486” in Korean news platforms and in English for the US news platforms.

-> (3rd Revision) The search included news articles that contained the keywords “Mifepristone,” “Misoprostol,” “Mifegyne,” and “RU-486” in either the title or the main context. On the Korean news platforms these keywords were searched in Korean, while English terms were used for the US news platforms.

[Location: Results section]

(2nd Revision) Following the text analysis pipeline described in Fig 1, we analyzed 7,938 Korean and 650 US news articles on medical abortion with mifepristone published between 2006 and 2022.

-> (3rd Revision) Following the text analysis pipeline described in Fig 1, duplicate articles were automatically removed using a pre-specified script, after which all remaining articles were manually reviewed; no further exclusions were required. A total of 7,938 Korean articles and 650 US articles on medical abortion with mifepristone, published between 2006 and 2022, were included in the final analysis.

[It has been added at the end of the manuscript]

(3rd Revision) Data Availability Statement

The datasets analyzed in this study were derived from publicly available online news articles retrieved from Google News and Naver News between January 2006 and December 2022. Both the processed datasets and the analysis code used in this study are available from the corresponding author upon reasonable request. However, the exact number of accessible articles or the composition of the corpora may vary in the future depending on changes in news outlet availability or platform crawling policies.

Reviewer’s Comment To further support interpretation, I suggest including a short summary section that explicitly highlights the key differences observed between the two countries in terms of themes and sentiments.

[Location: Discussion section]

(2nd Revision) This study …, we identified prominent topics, underlying sentiments, and word associations that illustrate public attitudes and policy discourse regarding medical abortion within regulatory and healthcare contexts.

-> (3rd Revision) This study …, we identified prominent topics, underlying sentiments, and word associations that illustrate public attitudes and policy discourse regarding medical abortion within regulatory and healthcare contexts.

The findings revealed media coverage in South Korea primarily emphasized the regulatory approval process, safety concerns, and social controversies. In contrast, the US media discourse was characterized by legal debates, clinical applications, and accessibility, with more polarized sentiment patterns.

Reviewer’s Comment In the Discussion, two aspects warrant careful attention:

Avoid implying causal relationships that are not empirically grounded. For example, the assumption that illegal online trade results directly from regulatory restrictions should be carefully qualified as a possible causal link or interpretative hypothesis, unless further evidence is provided.

The influence of media narratives on public opinion is mentioned but not sufficiently explored. A brief reflection on how and why media framing can shape public understanding—supported by relevant literature—would add theoretical depth and policy relevance to the discussion.

[Location: Discussion section]

(2nd Revision) Such reporting may reflect unintended consequences of stringent regulatory controls, inadvertently fostering unsafe practices and misuse and thereby raising public health and safety concerns.

-> (3rd Revision) Such reporting may suggest possible unintended associations between stringent regulatory controls and unsafe practices and misuse, raising public health and safety concerns [34].

(2nd Revision) This omission suggests a strong media focus on practical considerations related to the introduction of medical abortion rather than on legal debates surrounding the abolition of abortion laws.

-> (3rd Revision) This omission may indicate a strong media focus on practical considerations related to the introduction of medical abortion rather than on legal debates surrounding the abolition of abortion laws.

(2nd Revision) This practice may indicate the evolving nature of public discourse on abortion in South Korea; it merits a further investigation into how media framing shapes the public understanding of complex health concerns.

-> (3rd Revision) The media narratives are becoming increasingly important in shaping public belief and opinions, ultimately contributing to social change [14,35]. This practice may indicate the evolving nature of public discourse on abortion in South Korea, warranting further investigation into how media framing shapes the public understanding of complex health concerns.

Reviewer’s Comment The Conclusion successfully summarizes the findings but remains rather general. It would benefit from being more precise and action-oriented, particularly by offering concrete recommendations. For example, the authors could suggest how media coverage might adopt a more balanced and informative narrative regarding medical abortion, or how policymakers might engage with media discourses more strategically.

[Location: Conclusion section]

(2nd Revision) This study identifies ~ in each country. The topic modelling results showed that the Korean news focuses on regulatory challenges, safety concerns, and social perceptions, whereas the US coverage underscores clinical aspects and access-related debates.

Our findings reveal ~ surrounding medical abortion

-> (3rd Revision) This study identifies ~ in each country. The Korean news coverage focused on regulatory challenges, safety concerns, and social perceptions, whereas the US coverage underscored clinical aspects and access-related debates.

Our findings reveal ~ surrounding medical abortion through media discourses. Furthermore, it is essential for the media to adopt more balanced and informative narratives when addressing culturally and ethically sensitive topics, including medical abortion

---

## [Decision Letter · Decision Letter 3]

29 Jan 2026

Comparative analysis of social issues toward medical abortion using mifepristone in South Korea and the US: Topic modelling and sentiment analysis

PONE-D-25-00622R3

Dear Dr. Song,

We’re pleased to inform you that your manuscript has been judged scientifically suitable for publication and will be formally accepted for publication once it meets all outstanding technical requirements.

Kind regards,

Andrea Cioffi

Academic Editor

PLOS One

Additional Editor Comments (optional):

Reviewers' comments:

Reviewer's Responses to Questions

**Comments to the Author**

Reviewer #5: All comments have been addressed

2. Is the manuscript technically sound, and do the data support the conclusions?

Reviewer #5: Yes

3. Has the statistical analysis been performed appropriately and rigorously?

Reviewer #5: I Don't Know

4. Have the authors made all data underlying the findings in their manuscript fully available?

Reviewer #5: Yes

5. Is the manuscript presented in an intelligible fashion and written in standard English?

Reviewer #5: Yes

Reviewer #5: Thank you to the Editor for the opportunity to review the manuscript again. The authors have carefully addressed my revision suggestions and implemented the requested changes. Overall, in its current form, the paper has improved substantially and, in my view, could be considered suitable for publication.

**Do you want your identity to be public for this peer review?** For information about this choice, including consent withdrawal, please see our Privacy Policy

Reviewer #5: No

---

## [Editor Report · Acceptance letter]

PONE-D-25-00622R3

PLOS One

Dear Dr. Song,

I'm pleased to inform you that your manuscript has been deemed suitable for publication in PLOS One. Congratulations! Your manuscript is now being handed over to our production team.

Kind regards,

on behalf of

Dr. Andrea Cioffi

Academic Editor

PLOS One